# Photodegradation of Methylene Blue and Rhodamine B Using Laser-Synthesized ZnO Nanoparticles

**DOI:** 10.3390/ma13194357

**Published:** 2020-09-30

**Authors:** Damjan Blažeka, Julio Car, Nikola Klobučar, Andrea Jurov, Janez Zavašnik, Andrea Jagodar, Eva Kovačević, Nikša Krstulović

**Affiliations:** 1Institute of Physics, Bijenička cesta 46, 10000 Zagreb, Croatia; dblazeka@ifs.hr (D.B.); jcar@ifs.hr (J.C.); nklobucbbb@gmail.com (N.K.); 2Department of Gaseous Electronics, Jožef Stefan Institute, Jamova cesta 39, SI-1000 Ljubljana, Slovenia; andrea.jurov@ijs.si (A.J.); janez.zavasnik@ijs.si (J.Z.); 3Jožef Stefan International Postgraduate School, Jamova cesta 39, SI-1000 Ljubljana, Slovenia; 4GREMI, UMR7344 CNRS/Université d’Orléans, F-45067 Orléans, France; andrea.jagodar@univ-orleans.fr (A.J.); eva.kovacevic@univ-orleans.fr (E.K.)

**Keywords:** photocatalysis, pulsed laser ablation in water, ZnO nanoparticles, Methylene Blue, Rhodamine B

## Abstract

In this paper we examined the photocatalytic efficiency of a laser-synthesized colloidal solution of ZnO nanoparticles synthesized by laser ablation in water. The average size of the obtained colloidal ZnO nanoparticles is about 47 nm. As revealed by electron microscopy, other nanostructures were also present in the colloidal solution, especially nanosheets. A photocatalytic degradation of UV-irradiated Methylene Blue and Rhodamine B solutions of different concentration in the presence of different ZnO catalyst mass concentrations was studied in order to examine their influence on photodegradation rates. ZnO nanoparticles have shown high photocatalytic efficiency, which is limited due to different effects related to UV light transmittivity through the colloidal solution. Therefore, increasing catalyst concentration is effective way to increase photocatalytic efficiency up to some value where photodegradation rate saturation occurs. The photodegradation rate increases as the dye concentration decreases. These findings are important for water purification applications of laser-synthesized ZnO nanoparticles.

## 1. Introduction

Water pollution is one of the greatest ecological problems which is expected to become even larger in the future and occurs due to large amount of industrial waste material released into natural waters. Photodegradation of pollutants (organic or inorganic) into simpler and non-toxic components using photocatalytic materials is a very promising and widely researched technique for wastewater treatment [1]. Zinc oxide (ZnO) is one of the most examined and promising photocatalytic semiconductor materials after TiO_2_ due to its low cost, high photocatalytic efficiency and non-toxicity [2]. One of the advantages of ZnO over TiO_2_ is much larger electron mobility (200–300 cm^2^ V^−1^s^−1^ in ZnO compared to 0.1–4 cm^2^ V^−1^s^−1^ in TiO_2_) which contributes to larger photocatalytic efficiency in photodegradation of surrounding pollutants due to rapid electron transfer. On the other hand, the recombination rate of photogenerated electron-hole (e^−^/h^+^) pairs is also large, decreasing their availability for redox reactions with surrounding material and increasing energy dissipation as heat. The bandgap in ZnO (3.37 eV) is similar as in TiO_2_ (3.2 eV in anatase) and therefore can absorb only about 5% of sunlight radiation energy [3,4]. The main disadvantage of ZnO is the occurrence of photocorrosion during light irradiation that results in a fast decrease in photocatalytic activity during irradiation and low endurance of ZnO-based photocatalysts [5]. Photocorrosion mostly occurs due to reactions between holes and surface oxygen vacancies and the main approaches to overcome it include surface modification techniques [6]. These techniques mostly include formation of an additional layer with some of the following features: high intrinsic chemical stability, improved charge tranfer rate at the solid/liquid interface, improved e^−^/h^+^ separation (etc due to formation of Schottky junction electric field) or high hole-conductivity (etc p-type material) [7]. The most frequently used layers in ZnO photocorrosion prevention are carbon-based as graphite [8], fullerene [9], graphene [10], polyaniline [11]. A large advantage of ZnO over TiO_2_ in water treatment is its high antibacterial activity towards a broad range of bacteria (especially Gram-positive) which occurs even in dark conditions due to penetration through the bacterial membrane [12]. Mechanisms which lead to photodegradation of surrounding chemicals are similar to these which occur in TiO_2_. The bottom of the conduction band is at −0.5 V vs. the normal hydrogen electrode (NHE) which allows electron transfer to O_2_ (−0.33 V vs. NHE) producing very reactive superoxide $O_{2}^{-}$ and the top of valence band is at +2.7 V vs. NHE which allows hole transfer to H_2_O (2.53 V vs. NHE) producing another reactive radical OH*. These radicals are the main contributors to the degradation of chemical pollutants in water which are adsorbed at the catalyst surface and their production rate is proportional to the e^−^/h^+^ excitation rate in the photocatalytic material which occurs due to light absorption [3,13]. Noted values of ZnO redox potential are the same as in TiO_2_ [14]), that is the main reason for the large similarities in degradation mechanisms between these two materials. It is worth mention that both ZnO and TiO_2_ are great photocatalytic materials for CO_2_ gas photoreduction, especially in the presence of H_2_O, that can lead to the formation of useful products [15]. Some of the products formed in CO_2_ reduction processes are methane, carbon monoxide, methanol, formic acid, ethane, ethanol and oxalate. The first step in CO_2_ photoreduction is adsorption of CO_2_ molecule at catalyst surface, and then the mechanism $CO_{2}+e^{-}\to CO_{2}{}^{-}$ (where $e^{-}$ is photogenerated electron) needs to occur but due to the large potential barrier (−1.9 V vs. NHE), the reduction mechanism requires subsequent steps [16]. The large photocatalytic efficiency in degradation of carbonaceous contaminants (which occurs mainly due to CO_2_ adsorption) during UV-irradiation is well-established in the case of carbon-contaminated TiO_2_ catalysts [17].

In order to harvest a larger part of the solar irradiation spectrum, maximize the effectiveness and expand applicability of ZnO photocatalyst, visible-light driven photocatalysis is widely researched, with many possible solutions successfully applied. Metal doping (Ag, Mn, Cu, Co, Fe, Al) can lead to the creation of intra-band levels that lead to effective bandgap narrowing, which enables absorption of light with longer wavelengths, but at the same time the photocatalytic activity is deteriorated due to the fact that the new levels act as recombination centers. This problem is less present while doping ZnO with non-metals (C, N, S), which induces new states near the valence band [18,19,20,21]. Different types of heterogeneous nanostructures such as Janus nanoparticles [22] and plasmonic nanoparticles [23] can also show high visible-light catalytic activity. Another approach is optical light management (e.g., upconversion nanomaterials) which can be used for converting visible or NIR photons to UV photons [24].

Nanoparticles have large area/volume ratio, are great adsorbents and therefore are better for photocatalytic use than bulk materials. Nanoparticles have a large variety of applications in environmental issues [1]. ZnO nanoparticle (ZnONP) fabrication methods include sol-gel processing, vapour-liquid-solid technique, homogeneous and double-jet precipitation, hydrothermal synthesis, mechanical milling, organometallic synthesis, microwave method, spray pyrolysis, thermal evaporation, mechanochemical synthesis, [25,26] and laser synthesis which is used in this work [27]. We used pulsed laser ablation in liquid (PLAL) which is very promising method for the production of nanoparticles with high photocatalytic activity due to their high purity and is ecologically acceptable due to absence of any toxic byproducts. It also allows changing of broad range of parameters to influence nanoparticle shape, morphology and size [28,29,30]. More advanced laser techniques even allow laser-introduction of defects, bandgap and surface engineering on nanoparticles, which is a large opportunity for optimization of their photocatalytic properties [31].

In order to achieve optimization of NP colloidal solution photocatalytic activity, catalyst mass concentration is very important parameter. In this work, we measured the dependence of Methylene Blue (MB) and Rhodamine B (RB) photodegradation rates on various concentrations of laser-synthesized ZnO nanoparticles. MB or RB are the most frequently used dyes for measuring photodegradation rates in the presence of photocatalytic materials [32,33]. Information about photodegradation rate dependence on catalyst concentration can also help to optimize the parameters of laser ablation for the production of colloidal ZnONP (e.g., number of laser pulses). Part of this work relies on measuring photocatalytic activity dependence on initial dye concentration because from such information one can derive some assumptions about the applicability of laser-synthesized ZnONP in water pollution treatment, where average pollutant concentrations are usually much lower than the concentrations of organic dyes used in experiments.

## 2. Materials and Methods

### 2.1. Syntheses of the ZnO Nanoparticles

ZnONP colloidal solution was synthesized via a process of pulsed laser ablation of pure ZnO target (purity > 99.99%, GoodFellow, Huntingdon, UK) immersed in a glass beaker containing 25 mL of Milliq. water. The depth of water above the ZnO target was 2.5 cm. The experimental set-up is the same as we used in [27], with Nd:YAG laser (Quantel, Brilliant, Les Ulis, France) parameters as follows: number of pulses 10,000, wavelength 1064 nm, repetition rate 5 Hz, pulse energy 300 mJ (120 mJ delivered to the target), pulse duration 5 ns, fluence 79 J/cm^2^. Mass concentration of ablated ZnO material in as-synthesized colloid (in a form of nanoparticles and irregular material) was calculated from ablated crater-volume and known ZnO density. The crater volume was determined using an optical microscope (Leica DM2700M, Leica Microsystems, Wetzlar, Germany). As-synthesized colloidal solution is diluted in proper ratios to get solutions with 30%, 10% and 3% of initial ZnONP mass concentration. pH value of as prepared ZnONP colloidal solution is 7.3 ± 0.3. (after 30 min of UV irradiation it drops down to 6.9 ± 0.3).

### 2.2. Characterization

For structural characterization films of ZnO prepared by dropping of colloidal solution onto Si substrate until a visible film was formed. The shape, size and morphology of the as-synthesized ZnO nanoparticles or other ZnO structures which are present in synthesized solution were assessed by scanning electron microscope (SEM, JSM-7600F, Jeol Ltd., Tokyo, Japan), operating at 10 kV and utilizing Everhart-Thornley low-energy secondary electron detector. The fine structures were further analyzed by dual-beam scanning-electron microscope and focused ion beam (SEM-FIB, Helios NanoLab 650, FEI B.V., Eindhoven, The Netherlands), operating at 15 kV and additionally equipped with energy-dispersive X-ray spectrometer (X-max SDD, Oxford Instruments plc, Abingdon, UK).

ZnO film was structurally examined with GIXRD (grazing incidence X-ray diffraction) technique. The GIXRD measurements were carried out on a diffractometer equipped with a Co X-ray tube and a W/C multilayer for beam shaping and monochromatization (D5000, Siemens, Munich, Germany). The diffracted spectra were collected with a curved position sensitive detector (RADICON) in the angular range 2θ = 33°–88°. In all measurements, a fixed grazing incidence angle of α_i_ = 1.5° was used.

The X-ray photoelectron spectroscopy (XPS) analyses were carried out on the PHI-TFA XPS spectrometer (Physical Electronics Inc., Chanhassen, MN, USA). The analyzed area was 0.4 mm in diameter and the analyzed depth was about 3–5 nm. Sample surfaces were excited by X-ray radiation from monochromatic Al source at photon energy of 1486.6 eV. The high-energy resolution spectra were acquired with energy analyzer operating at resolution of about 0.6 eV and pass energy of 29 eV. During data processing the spectra from the surface were aligned by setting the C 1s peak at 284.8 eV, characteristic for C–C/C–H bonds. The accuracy of binding energies was about ±0.3 eV. Quantification of surface composition was performed from XPS peak intensities taking into account relative sensitivity factors provided by instrument manufacturer [34]. Two different XPS measurements were performed on each sample and average composition was calculated.

### 2.3. Photocatalytic Efficiency

A Hg UV lamp (homemade, UVA irradiation 285 mW/cm^2^, UVB irradiation 200 mV/cm^2^, UVC 8 mV/cm^2^) was used for irradiation of cuvettes which contain colloidal ZnONPs and organic dye (Methylene Blue or Rhodamine B) whose photodegradation rate was measured. Photoabsorbance measurements were made before irradiation and in steps of 10 min of irradiation in order to analyze gradual decrease of photoabsorption curve caused by photocatalytic degradation of dye using UV-Vis spectrophotometer (Lambda 25, Perkin Elmer, Waltham, MA, USA). Initial dye concentration was varied simply by adding different volumes of dye in non-irradiated ZnONP colloidal solution. At every ZnO catalyst mass concentration photocatalytic degradation rate for three different initial dye concentrations was measured—which in pure Milliq water have absorbance maxima *A*_0_ = 2, *A*_0_ = 1 and *A*_0_ = 0.5. MB has absorbance maximum at λ = 664 nm with corresponding extinction coefficient about 75,000 cm^−1^/M and RB has absorbance maximum at λ = 553 nm with corresponding extinction coefficient about 100,000 cm^−1^/M. Initial dye concentrations, calculated using Beer-Lambert law, were about 2.7 × 10^−5^ mol/L for MB, and about 2 × 10^−5^ mol/L for RB, both for *A*_0_ = 2.

## 3. Results and Discussion

### 3.1. Characterization of ZnO Colloidal Solution

#### 3.1.1. ZnO Concentration

The mass concentration of ZnO in as-synthesized colloidal solution is calculated according to a procedure described in [30].Volume of a crater $V_{ZnO}$ remaining on the ZnO target after ablation (or volume of ablated material) was determined from crater semi-profile shown in Figure 1. with a procedure described in [35]. In short, the crater has a Gaussian shape profile and it was determined by using an optical microscope taking images at different focal positions with respect to the target surface. From such images crater radii at certain depths were determined (whose values are shown in Figure 1).

Total crater volume $V_{ZnO}$ is obtained as a sum of frustums formed by consecutive radii at corresponding depths. Crater volume $V_{ZnO}$ is 5.61 × 10^7^ μm^3^, which multiplied with ZnO density $\rho_{ZnO}$ ($5.61\;g/cm^{3}$) gives the total mass $m_{ZnO}$ of ZnO in as-synthesized colloidal solution. Finally, ZnO mass concentration $C_{100\%}$ in as-synthesized ZnO solution is calculated by dividing $m_{ZnO}$ with volume of Milliq water $V_{liq}$ in vessel where target was immersed and laser-irradiated:(1)$$C_{100\%}=\frac{m_{ZnO}}{V_{liq}}=\frac{\rho_{ZnO}\times V_{ZnO}}{V_{liq}}=\frac{5.61\frac{g}{cm^{3}}\times 5.61\times 10^{7}\mu m^{3}}{25\;mL}=12.6\;mg/L$$

ZnO as-prepared solution is further diluted with Milliq water to get diluted ZnO colloidal solutions ($C_{30\%},\;C_{10\%},\;C_{3\%}$) to study dependence of photocatalytic degradation of MB and RB on different ZnO colloidal mass concentrations (including the pure dye cases).

#### 3.1.2. SEM Microscopy Observations

Secondary electron (SE) SEM micrographs are shown in Figure 2. In Figure 2a it can be seen that, beside well-defined nanoparticles, different irregular structures are formed during ablation. The largest part of analyzed sample area looks like that shown in Figure 2b, with prevailing spherical nanoparticles as dominant nanostructures. From SEM images a size-distribution of nanoparticles is determined and shown in Figure 3. Size-distribution can be fitted as log-normal with distribution maximum at diameter d=47±2 nm. Formation of certain amount of irregular structures upon laser synthesis of ZnONP make us unavailable to calculate precisely number concentrations of ZnONP but only mass concentration of all formed structures (equals to a mass of ablated materials). Further in text we refer to ZnONP but taking into account that also other structures participate in photodegradation reactions at certain level which we cannot distinguish.

Energy-dispersive X-ray spectroscopy (EDS) analysis revealed that ratio between zinc and oxygen atoms number is 52:48 as it can be seen from Appendix A. It points out that ZnO is dominant material formed in laser ablation while no significant impurities or other elements were found by this technique.

A detailed SE-SEM observation of surface morphology of ablated ZnO spherical nanostructures (Figure 4a) revealed structured surface. In Figure 4b large nanosheets can be seen, which contain many spherical nanoparticles trapped or adhered on their surface.

#### 3.1.3. XRD and XPS Analyses

The crystallinity of ZnONPs was examined by GIXRD measurements. The GIXRD spectrum, together with the reference patterns for the ZnO phase (JCPDS-ICDD card #36-1451), is shown in Figure 5 and it reveals that ZnONP are crystalline. Experimental GIXRD spectra show only the peaks characteristic for the hexagonal wurtzite phase of ZnO crystallites. In addition, the presence of all main ZnO diffraction peaks in the experimental curves indicate randomly oriented crystallites within the film (no preferential orientation).

High energy resolution spectra Zn 2p_3/2_ and O 1s are shown in Figure 6a,b, respectively. Zn 2p_3/2_ peak was fitted with one component peak at 1021.5 eV which corresponds to Zn^2+^ states in the ZnO lattice. The O 1s spectrum was fitted with two components. The peak at 531.6 eV (87% of relative intensity) may be assigned partially to chemisorbed OH or C–O groups and partially to oxygen deficient regions within the matrix of ZnO and therefore with concentration of oxygen vacancies [36]. OH or C–O groups probably originating from surface contamination during sample preparation under atmospheric conditions as it can be concluded from XPS depth profiles shown in Appendix A (C and O features are only present at sample surface). The contamination was also observed form XPS survey spectrum shown in Appendix A. Peak at 530.3 eV (13% of relative intensity) was assigned to O^2−^ oxide structure in ZnO lattice.

#### 3.1.4. Bandgap Determination

In order to calculate bandgap of as-synthesized colloidal ZnONP a Tauc plot was made from UV-Vis data as shown in Figure 7. A Tauc plot for was used direct bandgap calculation (${\left(Ahf\right)}^{2}\;vs.\;hf$) because ZnO is known as direct-bandgap semiconductor (shown in the inset in Figure 7). The direct-bandgap from the Tauc plot is 3.30 eV, which is very close to well-known value of the ZnO bandgap (3.37 eV). Larger bandgap deviations would point to changes and deformations in the optical and crystal properties in the ZnO material that could have a high impact on the physical and photocatalytic properties of colloidal solutions.

### 3.2. Photodegradation Rate Calculation

In Figure 8 an example of the decrease of absorbance of irradiated solution during photodegradation of (a) MB and (b) RB dyes is shown. These examples are made with solutions which have as-synthesized concentration of ZnO (mass concentration $C_{100\%}$) and initial amount of dye which corresponds to dye photoabsorbance maximum *A*_0_ = 2 (in an equivalent amount of pure water).

Appendix A shows the time-dependence of photoabsorbance maximum of UV-irradiated colloidal solution which contains (a) MB and (b) RB and four different amounts of ZnO catalyst mass concentrations in case of initial dye concentration corresponding to *A*_0_ = 2.

Photocatalytic degradation rates were calculated using pseudo-first order reaction rate model, described in [37,38], using the formula:(2)dC/dt=−DR×C
where *C* is the concentration of dye while *DR* is the rate constant or simply the degradation rate of the photocatalytic reaction. The solution of Equation (2) is given by:(3)$$C\left(t\right)=C_{0}e^{-DR\times t}$$
where $C_{0}$ is initial dye concentration and t is irradiation time.

From the Beer-Lambert law A=σ·C·l one can conclude that the dye time-dependent photoabsorbance peak A is proportional to the concentration of dye *C*, because the cross-section σ and absorption path-length l do not change during photodegradation. One can simply take *C*/*C*_0_ = *A*/*A*_0_. That is the reason one can use Equation (3) and photoabsorption measurements for determination of degradation rates. In Figure 9 plots of ln(*C*/*C*_0_) vs. *t* for (a) MB and (b) RB are shown as an example for the *A*_0_ = 2 case. From linear regression of ln(*C*/*C*_0_) vs. *t* a photodegradation rates *DR* and *DR* errors were evaluated.

Photodegradation half-time $t_{1/2}$ (time after the dye concentration drops to a half of its initial value) is also determined for the related reactions. In Table 1 the corresponding photodegradation rates, photodegradation half-times and $R^{2}$ from fit are summarized for MB and RB.

The dependence of photodegradation rates on ZnONP catalyst mass concentration (including the pure dye cases) is shown in Figure 10 for (a) MB and (b) RB for all three initial dye concentrations (*A*_0_ = 2, 1 and 0.5). *DR* errors are also shown for each *DR* value.

The MB photodegradation rate, shown in Figure 10a, grows while *C*_ZnO_ increases over the full measured range, but the slope of *DR* vs. *C*_ZnO_ curve decreases in the same range. Such a result can be explained in terms of the interplay between two effects. The first effect contributes to photodegradation rate growth when the amount of photocatalytic material increases and hence the area of the catalyst surface. It means that the number of active sites available for e^−^/h^+^ excitation and redox reactions is larger, so the production rate of reactive radicals responsible for photodegradation of dye increases. The second effect contributes to the photodegradation rate decrease due to decreasing of UV light transmittivity in the colloidal solution with increasing ZnO concentration. This means that e^−^/h^+^ excitation caused by UV light irradiating photocatalytic material will also decrease, especially in levels of solution which are at larger distance from the light incidence surface of the cuvette.

According to this analysis, it is expected that there exists some catalyst concentration *C*_M_ where the photodegradation rate achieves a maximal value due to interplay of the above- mentioned effects. The existence of such optimal concentration *C*_M_ can be seen in [39] while measuring MB photodegradation in the presence of different concentrations of TiO_2_ nanoparticles, in [40] when varying the ZnO and TiO_2_ catalyst load for photocatalytic degradation of Acid-brown 14, and in [41] when varying the ZrO_2_ catalyst concentration for photocatalytic degradation of Methylene Blue and Rhodamine B.

Catalyst concentrations examined in this work are obviously lower than *C*_M_, although saturation of MB photodegradation rate is obtained when increasing *C*_ZnO_. In our case, the highest photodegradation rate of MB was achieved when *A*_0_ = 0.5 and *C*_ZnO_ = 100% where *DR* = 0.0983 min^−1^ and *t*_1/2_ = 7.1 min. It is interesting to compare this photocatalytic efficiency with the one obtained using ZnO nanoparticles synthesized by other methods, for example sol-gel or precipitation. These methods were used for synthesis of ZnO nanoparticles in [42] where photodegradation of UV-irradiated MB in their presence was measured for different catalyst mass concentrations. Here, the optimal catalyst load was determined to be 250 mg/L, and corresponding photodegradation rates were 0.008 min^−1^ for ZnONP synthesized by sol-gel, and 0.0079 min^−1^ for ZnONP synthesized by percipitation method, both at initial MB concentration 20 mg/L. The degradation rates which we obtained are more than 10 times larger at about 20 times lower concentration of ZnO catalyst, which points to the great photocatalytic efficiency of the laser-synthesized ZnO nanoparticles in our work.

Photodegradation rate of RB shown in Figure 10b increases with ZnO concentration at all three applied dye concentrations, but the saturation is much less emphasized than in the MB case. It means that, in this case, the first effect mentioned above in the MB degradation analysis, related to changing of the active surface area of photocatalytic material, contributes much more than the second effect, related to the changing transmittivity of UV light. The reason for such a difference, when compared to the MB case, can be attributed to the much lower UV absorbance of the RB dye itself as it can be seen in Appendix A. Therefore, a UV light intensity which is sufficient for significant photocatalytic activation is present at deeper solution distances measured from the light-incidence side of the cuvette when compared to MB at the same ZnO concentration value. Higher values of *DR* error in MB degradation compared to RB could be attributed to the fact that during MB photodegradation much larger changes in UV transmittivity of the irradiated solution occur than in the RB case, that leads to more significant *DR* change during degradation time. Here, the highest photodegradation rate of RB was achieved when *A*_0_ = 0.5 and *C*_ZnO_ = 100% where *DR* = 0.0537 min^−1^ and *t*_1/2_ = 12.9 min.

In the case of MB, the difference between photodegradation rate at ZnO catalyst concentrations $C_{100\%}$ and $C_{30\%}$ is smaller than 20%. This means that colloidal solutions with ZnO catalyst concentrations close to $C_{30\%}\approx 4mg/L$ can be used in photocatalytic degradation of MB with almost the same effect, what consequently leads to a material and energy saving and minimization of possible ZnONP agglomeration.

In addition, a comparison between the photodegradation rates for *C*_100%_ and *C*_30%_ at different initial dye concentrations is given in Table 2. When observing the degradation rate dependence on initial dye concentration it is obvious that the photodegradation rate decreases with increasing initial dye concentration. Such a relation is also obtained in [32,40,43] where the explanation for such a result is attributed to several reasons.

The first one is UV-light absorption at dye molecules (both free or adsorbed at catalyst surface) and their degradation byproducts, so less light intensity is available for creation of e^−^/h^+^ pairs in nanoparticles. This effect should be larger in MB than in RB due to the high UV absorbance of MB (note that the MB absorption features around 300 nm coincide with both the emission line at 300 nm and emission band peaking at 280 nm from Hg lamp as shown in Appendix A), and this is probably why the ratio between photodegradation rates at different A_0_ for the same *C*_ZnO_ is larger in the MB than in the RB case. That can be seen in the last row of the Table 2 where the ratio *DR* (*A*_0_ = 0.5)/*DR* (*A*_0_ = 2) is given for MB and RB degradation in the presence of *C*_100%_ and *C*_30%_ ZnO catalyst concentrations.

Furthermore, excessive coverage of the catalyst surface with dye molecules can significantly decrease the degradation rate because the UV-screening effect prevents production of a sufficient amount of reactive radicals needed for immediate degradation of the adsorbed species. Therefore, the effective area of the active catalyst surface is decreased so it affects the photodegradation rate in the same way as decreasing the amount of photocatalyst.

The next reason is consumption of reactive radicals due to interactions with degradation byproducts, so less of them are available for dye degradation. This effect is enhanced due to fact that dye molecules should be adsorbed at the catalyst surface in order to be photodegraded, so degradation byproducts are produced very close to the active catalyst area and have a large probability to take part in reactions with radicals.

## 4. Conclusions

Colloidal solution synthesized by pulsed laser ablation of ZnO target, which contains mostly ZnO nanoparticles, has shown strong photocatalytic properties in the degradation of Methylene Blue and Rhodamine B. The photodegradation rate is about 40% higher for Methylene Blue when compared to Rhodamine B. Within the range of catalyst and dye concentrations used in this work, photocatalytic activity is larger for larger concentrations of ZnO catalyst and for smaller initial dye concentration. Degradation rate growth while increasing catalyst concentration is more emphasized in Rhodamine B than in Methylene Blue. For Methylene Blue, saturation of the degradation rate is obtained in the observed catalyst concentration range. These discrepancies are in relation with propagation of UV light through the medium which is affected by the ZnO concentration and the UV absorbance of the dye (Methylene Blue has a much stronger UV absorption than Rhodamine B). The fact that photodegradation rate is much larger at lower dye concentrations is very promising in terms of water purification applications, where pollutant concentrations are usually much lower than those used in this paper.

## Figures and Tables

**Figure 1 materials-13-04357-f001:**
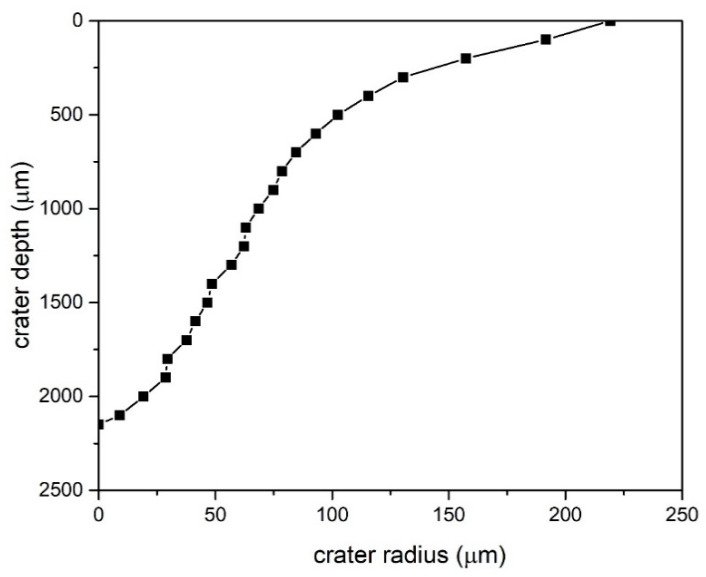
Semi-profile of a crater left after ablation in ZnO target (*x*–*y* axes are not to scale).

**Figure 2 materials-13-04357-f002:**
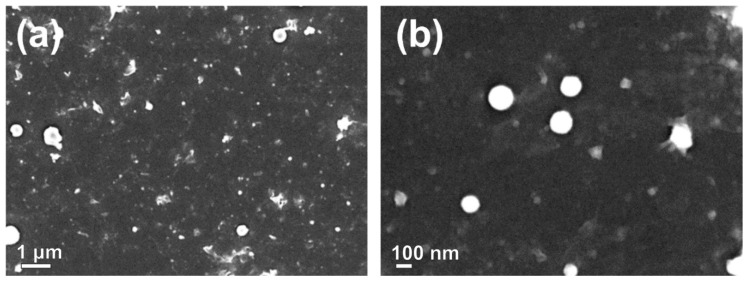
SE-SEM micrographs, (**a**) overview of synthesized nanoparticles and nanostructures and (**b**) detailed view of spherical ZnO nanoparticles.

**Figure 3 materials-13-04357-f003:**
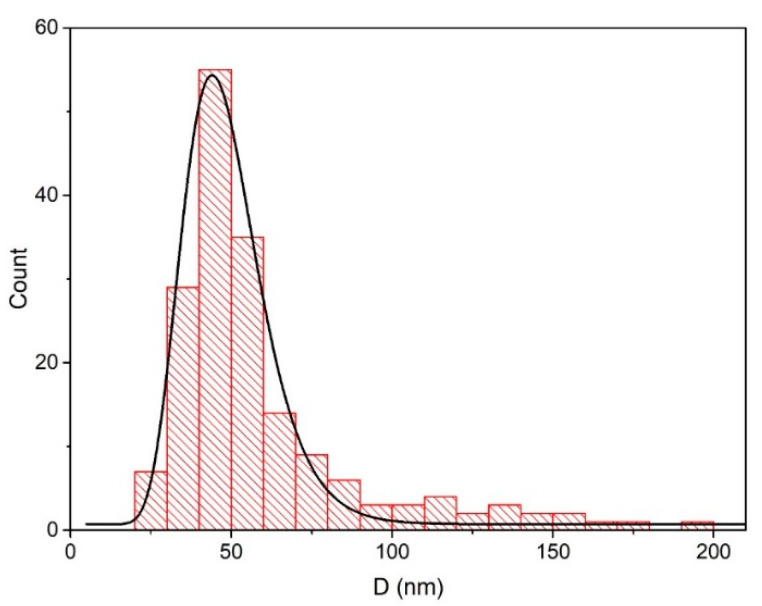
Size distribution of ZnO nanoparticles.

**Figure 4 materials-13-04357-f004:**
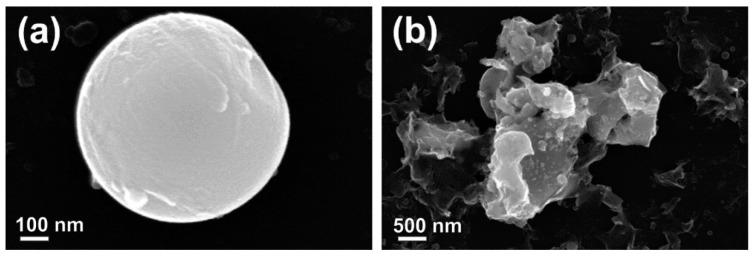
SE-SEM micrographs of (**a**) individual spherical ZnO nanoparticles showing structured morphology; and (**b**) large nano-sheets, covered by nano-sized particles.

**Figure 5 materials-13-04357-f005:**
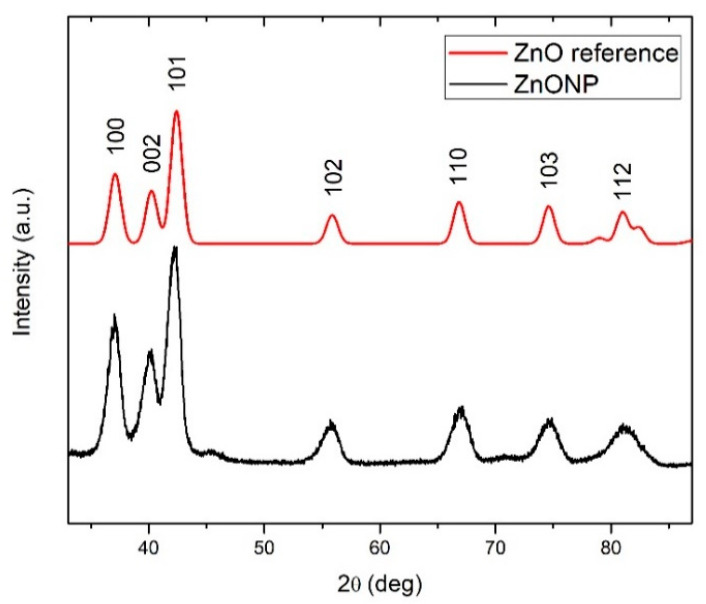
XRD pattern of ZnO.

**Figure 6 materials-13-04357-f006:**
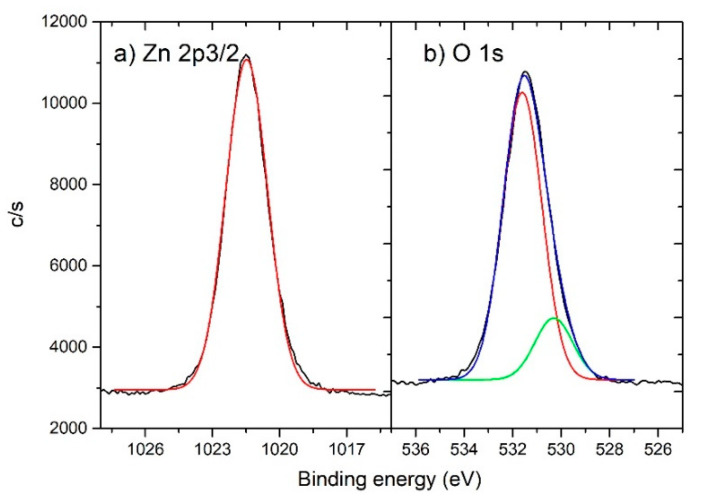
High resolution XPS spectra of ZnO with fit spectra for (**a**) Zn 2p_3/2_ and (**b**) O 1s.

**Figure 7 materials-13-04357-f007:**
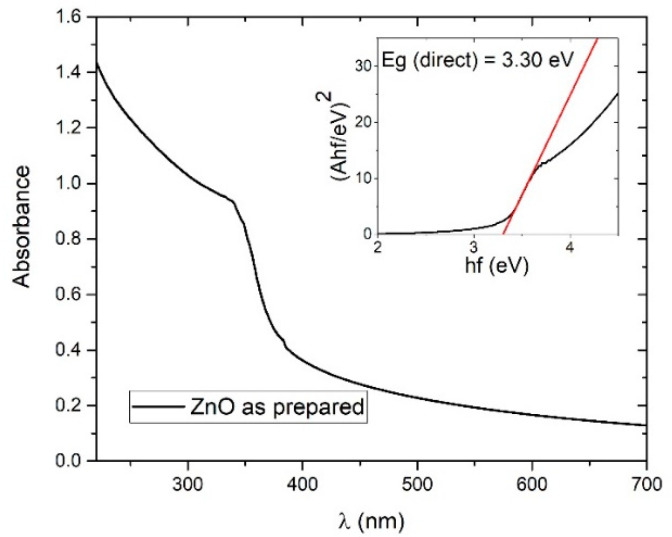
Photoabsorption spectrum of ZnO colloidal solution; Inset: Tauc plot for direct bandgap calculation.

**Figure 8 materials-13-04357-f008:**
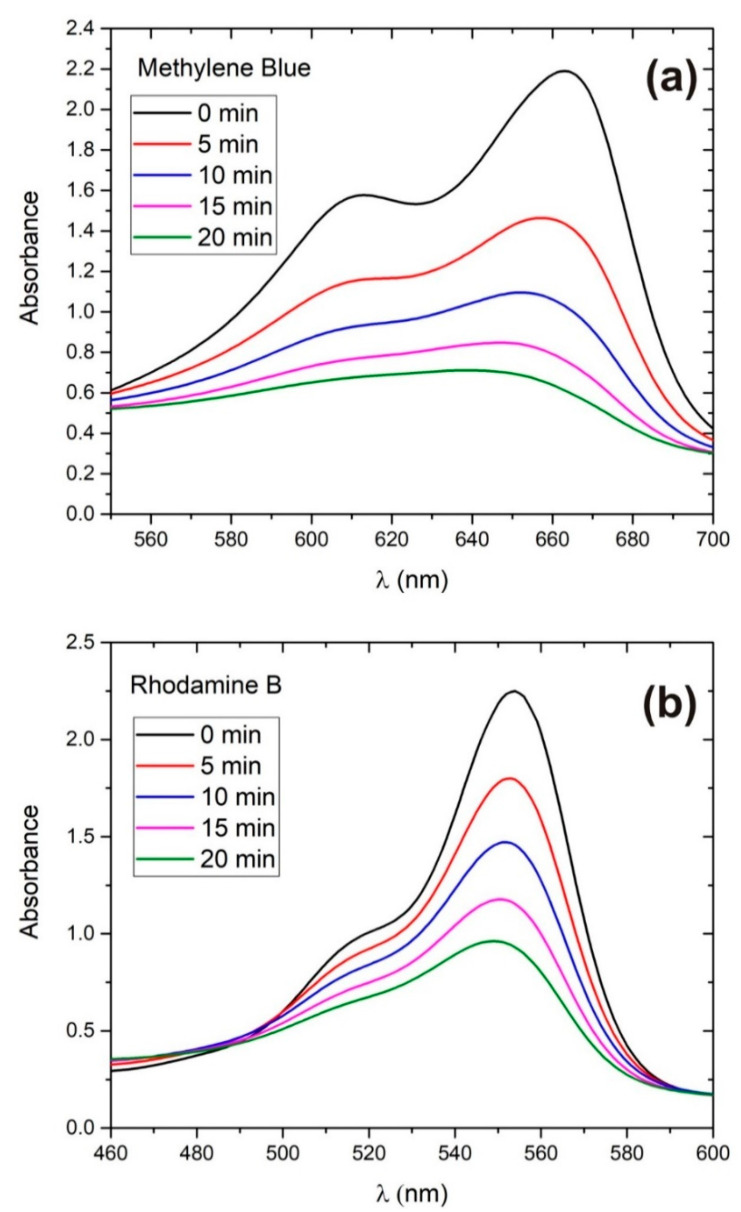
Photocatalytic degradation of (**a**) MB and (**b**) RB in as-prepared ZnO solution.

**Figure 9 materials-13-04357-f009:**
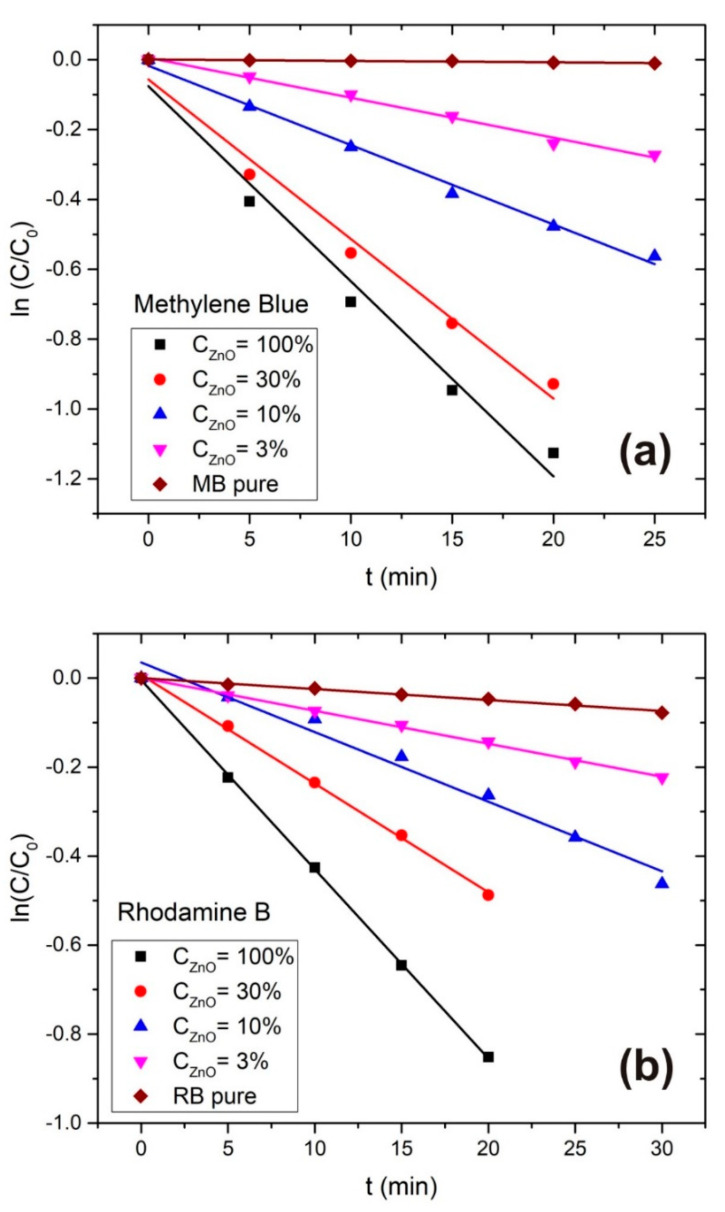
ln(*C*/*C*_0_) vs. t plot for (**a**) MB and (**b**) RB at different ZnONP concentrations.

**Figure 10 materials-13-04357-f010:**
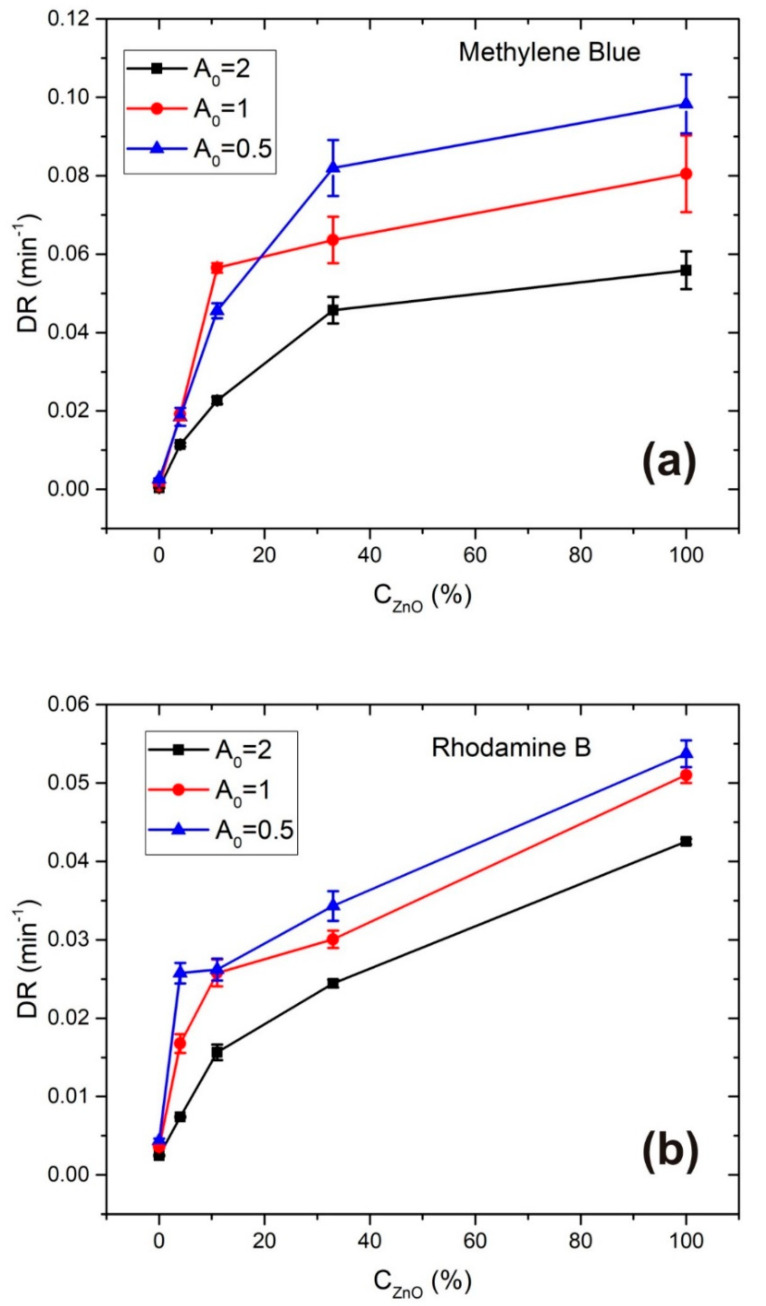
Photodegradation rates of (**a**) MB and (**b**) RB vs. ZnONP catalyst mass concentration for different initial dye concentrations.

**Table 1 materials-13-04357-t001:** Photodegradation rates, *R*^2^ and photodegradation half-time derived from pseudo-first order rate model for MB and RB for different concentrations of ZnONP and for different dye concentrations.

*A* _0_	*C* _ZnO_	MB			RB		
		*DR* (min^−1^)	*R* ^2^	*t*_1/2_ (min)	*DR* (min^−1^)	*R* ^2^	*t*_1/2_ (min)
2	*C* _100%_	0.0559	0.9784	12.4	0.0425	0.9998	16.3
	*C* _30%_	0.0457	0.9838	15.2	0.0244	0.9988	28.4
	*C* _10%_	0.0227	0.9934	30.5	0.0156	0.9793	44.3
	*C* _3%_	0.0114	0.9916	60.8	0.0074	0.9983	93.5
	pure dye	0.0004	0.9350	1733	0.0025	0.9899	281
1	*C* _100%_	0.0805	0.9855	8.6	0.0510	0.9855	13.6
	*C* _30%_	0.0636	0.9914	10.9	0.0301	0.9914	23.1
	*C* _10%_	0.0565	0.9991	12.3	0.0258	0.9991	26.9
	*C* _3%_	0.0192	0.9872	36.1	0.0168	0.9872	41.4
	pure dye	0.0010	0.9844	673	0.0036	0.9863	193
0.5	*C* _100%_	0.0983	0.9942	7.1	0.0538	0.9979	12.9
	*C* _30%_	0.0820	0.9925	8.5	0.0343	0.9942	20.2
	*C* _10%_	0.0456	0.9964	15.2	0.0262	0.9914	26.4
	*C* _3%_	0.0185	0.9709	37.5	0.0258	0.9927	26.9
	pure dye	0.0026	0.9688	266	0.0044	0.9800	159

**Table 2 materials-13-04357-t002:** Photodegradation rates of MB and RB for different initial dye concentrations and ZnO catalyst concentrations *C*_100%_ and *C*_30%_; Ratio *DR* (*A*_0_ = 0.5)/*DR* (*A*_0_ = 2) for each case.

Dye Abs. Maxima	*C* _100%_		*C* _30%_	
*A* _0_	*DR* (MB)/min^−1^	*DR* (RB)/min^−1^	*DR* (MB)/min^−1^	*DR* (RB)/min^−1^
2	0.056	0.043	0.046	0.024
1	0.081	0.051	0.064	0.030
0.5	0.098	0.054	0.082	0.034
*DR*(*A*_0_ = 0.5)/*DR* (*A*_0_ = 2)	1.75	1.26	1.78	1.42

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
