# Peer review of "Photodegradation of Methylene Blue and Rhodamine B Using Laser-Synthesized ZnO Nanoparticles"

_materials, 2020, doi:10.3390/ma13194357_

Round 1
Reviewer 1 Report
This paper deals with the laser-induced synthesis of ZnO nanoparticles for the photocatalysis application. Particularly, MR and RB photo-reductions were demonstrated under the UV irradiation. The demonstrated result is of interest, and the work scope is within the field of the journal. The novelty of this work is not high, but the provided experimental results well support the authors’ hypothesis. The followings should be addressed upon the resubmission of the revised version:
- Page 1, line 44 – According to the Porubaix diagram, ZnO is stable under pH around 10 by forming the Zn(OH)2 phase. ZnO is indeed corrodible under the light irradiance, but more detail information, for instance, pH or potential conditions should be provided since the ZnO is the main topic of discussion.
- Page 2, line 48 – The surface modification is definitely the most widely used approach in photocatalyst and PEC techniques, however, this needs to be more clearly described with relevant references. For instance, via adding additional layer forming, via surface kinetics engineering, etc. (10.1039/c6cs00918b).
- Page 3, line 140 – It would be great if the authors can provide how these values were obtained. Particularly, Vliq and Vzno.
- Page 6, line 182 – The XPS survey result should be provided. The survey data will be very useful to evaluate the purity of the sample.
- Page 8, line 208 – The absorption data using a blank sample is very important for verification of the functionality of the ZnO NP.
- Except for the points above, the relevant reference and brief photoreduction mechanism of the TiO2 should be provided along with the provided ZnO description. The TiO2’s mechanism for carbon contamination removal during the photocatalytic activities is very well established (10.1002/celc.201800918; 10.1021/am302903n).
Author Response
This paper deals with the laser-induced synthesis of ZnO nanoparticles for the photocatalysis application. Particularly, MR and RB photo-reductions were demonstrated under the UV irradiation. The demonstrated result is of interest, and the work scope is within the field of the journal. The novelty of this work is not high, but the provided experimental results well support the authors’ hypothesis. The followings should be addressed upon the resubmission of the revised version:
- Page 1, line 44 – According to the Porubaix diagram, ZnO is stable under pH around 10 by forming the Zn(OH)2 phase. ZnO is indeed corrodible under the light irradiance, but more detail information, for instance, pH or potential conditions should be provided since the ZnO is the main topic of discussion.
Answer: pH of pure ZnO colloid was measuerd before and after irradiation with UV light. It is shown that initial (untreated, as prepared) pH value of ZnO colloid is 7.3 while after 30 min of UV irradiation it drops to 6.9. It makes average pH value of 7.1±0.2. Here is graph (not included in supplemet material):
We added sentence in section 2.1:
„pH value of as prepared ZnONP colloidal solution is 7.3 ± 0.3. (after 30 min of UV irradiation it drops down to 6.9 ± 0.3).“
- Page 2, line 48 – The surface modification is definitely the most widely used approach in photocatalyst and PEC techniques, however, this needs to be more clearly described with relevant references. For instance, via adding additional layer forming, via surface kinetics engineering, etc. (10.1039/c6cs00918b).
Answer: The comment is added in Introduction section, including proposed reference [7] and we also adde new ones [8-11]:
„These techniques mostly include formation of additional layer with some of the following features: high intrinsical chemical stability, improved charge tranfer rate at the solid/liquid interface, improved e-/h+ separation (etc due to formation of Schottky junction electric field) or high hole-conductivity (etc p-type material) [7]. The most frequently used layers in ZnO photocorrosion prevention are carbon-based as graphite [8], fullerene [9], graphene [10], polyaniline [11].“
- Page 3, line 140 – It would be great if the authors can provide how these values were obtained. Particularly, Vliq and Vzno.
Answer: Vliq (=25mL) is a volume of a beaker where laser abltion takes place. We pointed out more precisely what Vliq is in section 3.1.1.
The description on how VznO is determined was added in 3.1.1.:
„In short, crater has Gaussian shape profile and it was determined by using optical microscope making images at different focal positions with respect to target surface. From such images crater radii at certain depths were determined (whose values are shown in Figure 1). Total crater volume VZnO is obtained as a sum of frustums formed by consecutive radii at corresponding depths.“
- Page 6, line 182 – The XPS survey result should be provided. The survey data will be very useful to evaluate the purity of the sample.
Answer: XPS survey spectrum is shown in added in supplement material as Figure S2. It shws contamination with C and O. The comment is added in section 3.1.3. after depth-profile analysis:
„OH or C-O groups probably originating from surface contamination during sample preparation under atmospheric conditions as it can be concluded from XPS depth profiles shown in Figure S1 (C and O features are only present at sample surface). The contamination was also observed form XPS survey spectrum shown in Figure S2.“
- Page 8, line 208 – The absorption data using a blank sample is very important for verification of the functionality of the ZnO NP.
Answer: UV degradation of pure MB and RB (no presence of ZnO) was measured and it was shown that degradation rates are very small especially for MB. Measured data were added in Figure 9 a) and b). and Figure 10 a) and b). The analysis is extended for those cases in Result section.
- Except for the points above, the relevant reference and brief photoreduction mechanism of the TiO2 should be provided along with the provided ZnO description. The TiO2’s mechanism for carbon contamination removal during the photocatalytic activities is very well established (10.1002/celc.201800918; 10.1021/am302903n).
Answer: The comment is added in Introduction section:
„Noted values of ZnO redox potential are the same as in TiO2 [14]), that is the main reason of large similarities in degradation mechanisms between these two materials. It is worth to mention that both ZnO and TiO2 are great photocatalytic materials for CO2 gas photoreduction, especially in the presence of H2O, that can lead to formation of useful products [15]. Some of the products formed in CO2 reduction processes are: methane, carbon monoxide, methanol, formic acid, ethane, ethanol and oxalate. The first step in CO2 photoreduction is adsorption of CO2 molecule at catalyst surface, and then the mechanism 〖CO〗_2+e^-→〖〖CO〗_2〗^- (where e^- is photogenerated electron) needs to occur but due to large potential barrier (-1.9 V vs. NHE), reduction mechanism acquires subsequent steps. [16]. The large photocatalytic efficiency in degradation of carbonaceous contaminants (which occurs mainly due to CO2 adsorption) during UV-irradiation is well-established in the case of carbon-contaminated TiO2 catalyst [17].“

Reviewer 2 Report
The manuscript by Blažeka and Co-authors, studied the photocatalytic activity of newly synthesized ZnO nanoparticles for the environmental applications. The study is interesting in the field nanotechnology. Synthesized nanoparticles were thoroughly characterized by using various techniques. Moreover, ZnO nanoparticles showed promising activity for dye degradation rhodamine B and methylene, at low concentrations. Overall the presented results and writing of the manuscript is constructive and focused. But, introduction section needs to be strengthened by adding some references in the first paragraph.
I would like to recommend adding this reference in the introduction section (Pathakoti, K.; Manubolu, M.; Hwang, H.-M. Chapter 48-Nanotechnology applications for environmental industry. In Handbook of Nanomaterials for Industrial Applications; Hussain, C.M., Ed.; Elsevier: Cambridge, MA, USA, 2018; pp. 894-907).
Minor comments:
Title needs to be revised:
Photodegradation of Methylene Blue and Rhodamine B using laser-synthesized ZnO nanoparticles
Author Response
The manuscript by Blažeka and Co-authors, studied the photocatalytic activity of newly synthesized ZnO nanoparticles for the environmental applications. The study is interesting in the field nanotechnology. Synthesized nanoparticles were thoroughly characterized by using various techniques. Moreover, ZnO nanoparticles showed promising activity for dye degradation rhodamine B and methylene, at low concentrations. Overall the presented results and writing of the manuscript is constructive and focused. But, introduction section needs to be strengthened by adding some references in the first paragraph.
I would like to recommend adding this reference in the introduction section (Pathakoti, K.; Manubolu, M.; Hwang, H.-M. Chapter 48-Nanotechnology applications for environmental industry. In Handbook of Nanomaterials for Industrial Applications; Hussain, C.M., Ed.; Elsevier: Cambridge, MA, USA, 2018; pp. 894-907).
Answer: That references is added as [1].
Minor comments:
Title needs to be revised:
Photodegradation of Methylene Blue and Rhodamine B using laser-synthesized ZnO nanoparticles
Answer: Title changed.

Reviewer 3 Report
The manuscript entitled "Photodegradation rate dependence on concentration of laser-synthesized ZnO nanoparticles as photocatalyst", reports the results obtained from a research in which ZnO nanoparticles are synthesized, by ablation technique. The photodegradative abilities of these nanoparticles were subsequently analyzed for methylene blue and rhodamine solutions. The synthesized ZnO nanoparticles have been characterized by different techniques such as XRD, XPS, electron microscopy analysis. Finally, the photo-catalytic reactions were studied as a function of the concentration of the nanoparticles.
The subject of the manuscript is consistent with the purpose of the journal and although the results obtained are not particularly original, it can be an element of study for researchers working in the field of photocatalysis and nanomaterials.
In light of the above, I believe that the manuscript can be accepted for its publication but after a major revision.
Here are some suggestions for authors:
- Paragraph 2.3. This part should specify if there are two solutions of Methylene Blue and Rhodamine or if it is a single solution containing both dyes.
- Paragraph 3.1.1. It is necessary to explain in more detail how it was obtained: "the volume of the crater is 5.61 * 107 mm3";
-Fig. 1. How were the data reported in figure 1 obtained?
- Fig. 1. The caption should be clearer
-Fig. 5 The y axis shows the values, although the intensity is in arbitrary units.
-Fig. 7/8. Are the figures reported in arbitrary units?
-Table 1. Why are only the data referred to A° = 2 shown in table 1? Perhaps it might be interesting to report the values for A° = 1 and 0.5.
Conclusion. The conclusions and the critical analysis of the results are a bit obvious. I suggest a deeper study of the data obtained. A further experimental study or for example a comparison with literature data referring to other types of nanoparticles could be useful.
Author Response
The manuscript entitled "Photodegradation rate dependence on concentration of laser-synthesized ZnO nanoparticles as photocatalyst", reports the results obtained from a research in which ZnO nanoparticles are synthesized, by ablation technique. The photodegradative abilities of these nanoparticles were subsequently analyzed for methylene blue and rhodamine solutions. The synthesized ZnO nanoparticles have been characterized by different techniques such as XRD, XPS, electron microscopy analysis. Finally, the photo-catalytic reactions were studied as a function of the concentration of the nanoparticles.
The subject of the manuscript is consistent with the purpose of the journal and although the results obtained are not particularly original, it can be an element of study for researchers working in the field of photocatalysis and nanomaterials.
In light of the above, I believe that the manuscript can be accepted for its publication but after a major revision.
Here are some suggestions for authors:
- Paragraph 2.3. This part should specify if there are two solutions of Methylene Blue and Rhodamine or if it is a single solution containing both dyes.
Answer: We made it clear.
- Paragraph 3.1.1. It is necessary to explain in more detail how it was obtained: "the volume of the crater is 5.61 * 107 mm3";
Answer: Now it it expalined in text how VznO is determine in more details in section 3.1.1.
We added explanantion: „In short, crater has Gaussian shape profile and it was determined by using optical microscope making images at different focal positions with respect to target surface. From such images crater radii at certain depths were determined (whose values are shown in Figure 1). Total crater volume V_ZnO is obtained as a sum of frustums formed by consecutive radii at corresponding depths.“
-Fig. 1. How were the data reported in figure 1 obtained?
Answer: Please see comment above.
- Fig. 1. The caption should be clearer
Answer: We changed caption to:
„Semi-profile of a crater left after ablation in ZnO target (x-y axes are not to scale).“
-Fig. 5 The y axis shows the values, although the intensity is in arbitrary units.
Answer: Values were removed.
-Fig. 7/8. Are the figures reported in arbitrary units?
Answer: We removed units from y-axiy as absorbance is dimensionless physical quantity.
-Table 1. Why are only the data referred to A° = 2 shown in table 1? Perhaps it might be interesting to report the values for A° = 1 and 0.5.
Answer: We added cases for A=1 and A=0.5.
Conclusion. The conclusions and the critical analysis of the results are a bit obvious. I suggest a deeper study of the data obtained. A further experimental study or for example a comparison with literature data referring to other types of nanoparticles could be useful.
Answer: In Results and discussion section we added comparison of photocatalitic activity with ZnONP synthesized with sol-gel and precipitation method.

Reviewer 4 Report
In this study, the authors prepared a colloidal solution of ZnO structures using a laser ablation method. Prepared ZnO structures at different concentrations were used for photocatalytic degradation of organic dyes. In general, the article is interesting and can be considered for publication after a revision process.
1) Experimental part. It is not clear what kind of vessel was used for experiments. Depending on the diameter (narrow or wide) the laser beam should undergo different paths in solution. Please describe it more clearly.
2) EDX was mentioned in the text but the data was not provided.
3) XRD analysis, provide an internationally accepted XRD reference (JCPDS, ISDD, etc.) and associated crystal parameters.
4) Photodegradation rate. Dyes can be self-degradated under UV light. Self-degradation data must be provided and excluded from the total degradation % within a given period of time.
5) It will be interesting to see the reusability of these samples. Can we reuse them after several cycles with the same degradation efficiency?
6) In general, statistical data was not provided. It is mandatory, especially for degradation experiments.
7) Introduction part. I suggest to slightly expand the discussion on how one can develop visible-light photocatalysts. The authors mentioned not all of them. In particular, doping, plasmonic effects, attachments were slightly mentioned. However, one can also consider Janus type particles and optical light management as well. See for example these works: DOI: 10.1063/1.5039926 and DOI: 10.1007/s11706-019-0482-z
Author Response
In this study, the authors prepared a colloidal solution of ZnO structures using a laser ablation method. Prepared ZnO structures at different concentrations were used for photocatalytic degradation of organic dyes. In general, the article is interesting and can be considered for publication after a revision process.
1) Experimental part. It is not clear what kind of vessel was used for experiments. Depending on the diameter (narrow or wide) the laser beam should undergo different paths in solution. Please describe it more clearly.
Answer: We made it more clearly in section 2.1.
2) EDX was mentioned in the text but the data was not provided.
Answer: Data are now provided in supplemet material as Figure S1.
3) XRD analysis, provide an internationally accepted XRD reference (JCPDS, ISDD, etc.) and associated crystal parameters.
Answer: We added reference JCPDS-ICDD card #36-1451
4) Photodegradation rate. Dyes can be self-degradated under UV light. Self-degradation data must be provided and excluded from the total degradation % within a given period of time.
Answer: UV degradation of pure MB and RB (no presence of ZnO) was measured and it was shown that degradation rates are very small especially for MB. Measured data were added in Figure 9 a) and b). and Figure 10 a) and b). The analysis is extended for those cases in Result section. Because degradation rates for pure MB and RB are very small it would not change results rates if we exclude them from degradation rates in presence of catalyst. Anyway, we think that it is not necessarry to exclude them but only compare them with the case when catalyst is present as we measure photocatalysis of whole system.
5) It will be interesting to see the reusability of these samples. Can we reuse them after several cycles with the same degradation efficiency?
Answer: We can not reuse the samples as ZnONP precipitate and are covered with MB or RB degradation products so the reproducibility will fail. We think that they will work up to some level but never match original performance. To consider possibility to reuse such samples more further analysis is required due to change of related parameters with respect to initial conditions.
6) In general, statistical data was not provided. It is mandatory, especially for degradation experiments.
Answer: Statistical errors were added in Figure 10. Errors are shortly discussed in text in section Results and discussion too.
7) Introduction part. I suggest to slightly expand the discussion on how one can develop visible-light photocatalysts. The authors mentioned not all of them. In particular, doping, plasmonic effects, attachments were slightly mentioned. However, one can also consider Janus type particles and optical light management as well. See for example these works: DOI: 10.1063/1.5039926 and DOI: 10.1007/s11706-019-0482-z
Answer: References are added together with following text (in Introduction section):
„In order to harvest larger part of solar irradiation, maximize the effectiveness and expand applicability of ZnO photocatalyst, the visible-light driven photocatalysis is widely researched, with many possible solutions successfully applied. Metal doping (Ag, Mn, Cu, Co, Fe, Al) can lead to creation of intra-band levels that leads to effective bandgap narrowing, which enables absorption of light with longer wavelengths, but at the same time the photocatalytic activity is deteriorated due to the fact that new levels act as recombination centers. This problem is less present while doping ZnO with non-metals (C, N, S), which induces new states near the valence band [18-21]. Different types of heterogeneous nanostructures as Janus nanoparticles [22] and plasmonic nanoparticles [23] also can show high visible-light catalytic activity. Another approach is optical light management (etc upconversion nanomaterials) which can be used for converting visible or NIR photons to UV photons [24].“

Round 2
Reviewer 3 Report
The manuscript has been adequately revised and I believe that in the present version it can be considered for its publication
Reviewer 4 Report
No more comments, a revised manuscript can be accepted for publication